# Disturbed Antioxidant Capacity in Patients with Systemic Sclerosis Associates with Lung and Gastrointestinal Symptoms

**DOI:** 10.3390/biomedicines11082110

**Published:** 2023-07-26

**Authors:** Neža Brezovec, Katja Perdan-Pirkmajer, Blaž Burja, Žiga Rotar, Joško Osredkar, Snežna Sodin-Šemrl, Katja Lakota, Saša Čučnik

**Affiliations:** 1Department of Rheumatology, University Medical Centre Ljubljana, 1000 Ljubljana, Slovenia; katja.perdanpirkamjer@kclj.si (K.P.-P.); blaz.burja@gmail.com (B.B.); ziga.rotar@kclj.si (Ž.R.); ssodin1@yahoo.com (S.S.-Š.); katja.lakota@kclj.si (K.L.); 2Faculty of Medicine, University of Ljubljana, 1000 Ljubljana, Slovenia; 3Institute of Clinical Chemistry and Biochemistry, University Medical Centre Ljubljana, 1000 Ljubljana, Slovenia; josko.osredkar@kclj.si; 4Faculty of Pharmacy, University of Ljubljana, 1000 Ljubljana, Slovenia; 5Faculty of Mathematics, Natural Sciences and Information Technologies, University of Primorska, 6000 Koper, Slovenia

**Keywords:** systemic sclerosis, oxidative stress, reactive oxidative metabolites, antioxidants, lipid peroxidation, DNA damage, DNA breaks

## Abstract

The correct balance between reactive oxygen species and antioxidant defense in an organism is disturbed in oxidative stress. To assess oxidative balance in 36 SSc patients and 26 healthy controls (HCs), we measured reactive oxidative metabolites (ROMs), total antioxidant capacity (TAC), lipid peroxidation (measuring 4-HNE), and DNA oxidative damage (measuring 8-OHdG) in serum. Furthermore, DNA breaks in leukocytes of 35 SSc patients and 32 HCs were evaluated using COMET. While we report high ROMs for both SSc patients and age/sex matched HC samples, there was a significant increase in TAC in SSc patients as compared to HCs, and thus also a significantly higher oxidative stress index in SSc patients. TAC was significantly higher in SSc patients with ILD and gastrointestinal involvement, as well as in patients with anti-topoisomerase antibodies. We observe no difference in serum lipid peroxidation status or oxidative DNA damage. However, SSc patients had significantly more leukocyte DNA breaks than HCs; the most damage was observed in patients treated with immunosuppressives. Thus, our study confirms presence of oxidative stress and increased DNA damage in leukocytes of SSc patients; however, it points toward increased antioxidant capacity, which needs to be further studied.

## 1. Introduction

Oxidative stress as an imbalance between oxidant and antioxidant components is present in systemic sclerosis (SSc) and plays an important role in the pathogenesis of the disease. Increased reactive oxidative metabolites (ROMs) have been described as appearing not only in the circulation and in the skin, but also at the level of individual cells such as fibroblasts, monocytes, T lymphocytes, and erythrocytes of patients with SSc [1,2]. ROMs are involved in the three main pathological processes of SSc: inflammation (macrophage polarization and NRLP3 inflammasome activation), autoimmunity (autoantibody induction, differentiation, and activation of B and T lymphocytes), and fibrosis (activation of fibroblasts, transforming growth factor-β (TGF-β), and matrix metalloproteinases). The main source of ROMs in SSc is thought to be tissue ischemia/reperfusion, manifesting as, e.g., Raynaud phenomenon. In addition, cytokines and growth factors such as interleukin (IL)-6, IL-3, tumor necrosis factor-α (TNF-α), platelet-derived growth factor (PDGF), and TGF-β, which are elevated in SSc, also promote the formation of ROMs. On the other hand, SSc is also characterized by a poorly functioning antioxidant system. Total antioxidant capacity (TAC), as well as individual antioxidants such as superoxide dismutase, catalase, vitamin C, vitamin E, and thiol were shown to be altered in SSc [3,4]. In addition, impaired function of the nuclear factor NRF2, considered the master regulator of inducible antioxidant responses, has been described and associated with fibrosis and inflammation [5], and several NRF2 activators have been investigated in clinical trials [6]. Oxidative stress in the organism can lead to oxidative damage of biological molecules—proteins, lipids, and nucleic acids. In SSc, systemic oxidative damage of lipids and DNA [1], as well as DNA damage at the cellular level and impaired mechanisms for DNA damage repair, have already been described [7,8]. DNA breaks have been shown to activate nuclear factor κB (NFκB) and type I interferon [9], a cytokine that plays an important role in the pathogenesis of SSc [10]. Autoantibodies against proteins of DNA repair mechanisms (Ku and topoisomerase I) are also frequently found in the blood of SSc patients [11].

Oxidative stress plays an important role in SSc by triggering inflammation and autoantibody production. However, research in this area is still quite sparse, especially regarding its association with clinical symptoms and the influence of therapy. Therefore, our aim was to investigate oxidative balance (ROMs and TAC) and to determine whether disturbed oxidative balance is also reflected in systemic oxidative lipid and DNA damage, as well as DNA breaks in leukocytes from SSc patients. In addition, we aimed to investigate the correlations of oxidative damage with clinical data and therapy to highlight potential consequences of oxidative imbalance for SSc patients.

## 2. Materials and Methods

### 2.1. Study Design, Patients, and Controls

The study, which was approved by the National Ethics Committee (0120-31/2018/4 and 0120-195/2021/6), included two separate cohorts of SSc patients and two separate cohorts of age- and sex-matched healthy controls (HCs), collected at different time points after signing informed consent forms. In the first part of the study, we examined systemic markers of oxidative stress in 36 SSc patients and 26 HCs. In the second part, we examined DNA damage in leukocytes. We enrolled 35 SSc patients and 32 HCs to study DNA damage at baseline, and samples of 22 SSc patients and 29 HCs were further evaluated for DNA damage after induced oxidative stress and DNA damage repair. The HC group was age- and sex-matched and included those who had no autoimmune rheumatic disease and no acute illness.

SSc patients were treated at the Department of Rheumatology, University Medical Centre Ljubljana. They all met the 2013 ACR/EULAR classification criteria for SSc [12] and signed informed consent. Patients were classified into limited cutaneous or diffuse cutaneous subgroups according to Le Roy’s criteria [13]. Clinical data were collected, including the presence of Raynaud’s phenomenon, digital ulcers, pitting scars, telangiectasias, calcinosis, and capillaroscopy results. Interstitial lung disease was diagnosed using high-resolution computed tomography, and pulmonary arterial hypertension was diagnosed using heart ultrasound and NT-proBNP measurements; for those with high-risk results, right heart catheterization was performed. Pulmonary function tests were performed to determine forced vital capacity (FVC), forced expiratory volume in 1 s (FEV1), and diffusing capacity for carbon monoxide (DL_CO_). Gastrointestinal tract (GIT) involvement was assessed clinically, and when indicated endoscopy was performed for confirmation. Blood tests were performed to measure erythrocyte sedimentation rate (ESR), C-reactive protein (CRP), complete blood count, and interleukin 6 (IL-6). The presence of autoantibodies such as anti-centromere antibodies (ACAs) and anti-topoisomerase I antibodies (ATAs) was determined in the patients’ sera using the in-house method for anti-extractable nuclear antigens [14] and the HEp-2 assay for antinuclear antibodies (Immuno-concepts, Sacramento, CA, USA). Patients’ serum negative for ACA and ATA were additionally tested for other SSc-related antibodies (Scl-70, CENP-A, CENP-B, PM-Scl 100, PM-Scl 75, Ku, RNA P III, U1-RNP, Th/To, Fibrilarin, NOR-90, and SS-A/Ro-52kD) using BlueDiver Dot Scleroderma12 IgG Immunodot kit (D-tek, Mons, Belgium). Data on patients’ pharmacotherapy (immunosuppressants, calcium channel blockers, prostacyclins, phosphodiesterase type 5 inhibitors, and analgesics) were also collected.

### 2.2. Measurement of Oxidative Stress Markers in the Systemic Circulation

Peripheral blood was collected into serum tubes, allowed to stand at room temperature for 30 min, and centrifuged (2000× *g* RCF, 10 min, room temperature) to separate the serum, which was then aliquoted and frozen at −80 °C. The oxidation status of the serum was determined using the d-ROM assay (measuring ROOH concentration), and the antioxidant potential was determined using the PAT assay (based on the ability to reduce Fe^3+^ to Fe^2+^). We used the REDOX fast kit (Innovatics Laboratories, Philadelphia, PA, USA) and the FRAS5 system (Innovatics Laboratories, Philadelphia, PA, USA). d-ROM: optimal range 250–300, borderline range 301–320, mild oxidative stress 321–340, moderate oxidative stress 341–400, high oxidative stress 401–500, very high oxidative stress > 500 U. Carr. TAC: antioxidant deficiency < 1800, mild deficiency 1800–2000, borderline range 2000–2200, normal range 2200–2800, very high > 2800 U. Cor. Based on the measured ROMs and TAC values, the oxidative stress index (OSI) was automatically determined from both measured values and converted to full numbers for ease of interpretation. Lipid peroxidation was assessed using 4-HNE in serum and DNA damage using 8-OHdG in serum using enzyme-linked immunosorbent assay (ELISA) kits according to the manufacturers’ instructions (4-HNE, MyBioSource, San Diego, CA, USA; 8-OHdG, Bio-Connect, Oss, The Netherlands).

### 2.3. COMET Assay

Peripheral blood was collected in heparin tubes (BD Vacutainer, Franklin Lakes, NJ, USA). The COMET assay was performed as previously described [15]. Briefly, 36 µL of blood was resuspended in 564 µL of 0.67% low-melt agarose (Certified Low-Melt Agarose, Bio-Rad Laboratories, Madrid, Spain) at 37 °C. An amount of 100 µL of the cell suspension was layered onto 6 slides using coverslips then cooled at 4 °C for 10 min, and the coverslips removed. Two slides were immediately covered with an additional layer of 0.5% low melt agarose, cooled at 4 °C for 10 min and placed overnight in alkaline lysis buffer (2 M NaCl (Merck, Darmstadt, Germany), 89 mM EDTA (Merck, Darmstadt, Germany), 8.9 mM Tris (Kemika, Zagreb, Croatia), 10% DMSO (CARLO ERBA Reagents, Cornaredo, Italy), and 1% Triton X-100 (Sigma, Darmstadt, Germany)). Four slides were treated with 3% H_2_O_2_ (Lekarna UKC, Ljubljana, Slovenija) at 37 °C and then rinsed with PBS. Two slides were then placed in the lysis buffer as previously described, while the remaining two slides were first covered with a coverslip and incubated at 37 °C to allow DNA reparation and then placed in the lysis buffer. The next day, the slides were immersed in alkaline electrophoresis buffer (0.3 M NaOH (Kemika, Zagreb, Croatia), 1 mM EDTA) for 30 min, followed by electrophoresis (30 min, 25 V, 300 mA). The slides were then rinsed twice with neutralization buffer (1.2 M Tris, pH = 7.5) for 10 min and with water. The cells on the slides were then stained with SYBR GreenⅠ nucleic acid gel stain (Invitrogen, Waltham, MA, USA) and examined under a fluorescence microscope at 400× magnification (Nikon eclipse TE300 and Axio Imager Z1, Carl Zeiss, Oberkochen, Germany). An amount of 100 cells per slide were classified into categories A–E based on tail length/shape (A = 0; B = 1; C = 2; D = 3; and E = 4), with category A representing intact DNA, and category E representing completely damaged DNA. The result was reported as a COMET score (sum of the percentage of cells in each category multiplied by the numerical value of the category). The higher the COMET score, the more DNA breaks were present. To avoid bias and error, the assay was performed in duplicate, always including both a patient sample and an HC sample in the same analysis. DNA damage was assessed in a blind fashion. For each condition, the average of the two duplicate slides was calculated.

### 2.4. Statistical Analysis and Clinical Significance

The normality of the Gaussian distribution was tested using the Shapiro–Wilk test. On this basis, statistical analysis of the data was performed with the two tailed Student *t* test or the Mann–Whitney U test for the comparison of 2 continuous variables and the ANOVA or the Kruskal–Wallis test when more than 2 variables were compared. Graphpad Prism v9.4.1 (GraphPad Software, San Diego, CA, USA) and IBM SPSS Statistics for Windows, v22 (IBM, Armonk, NY, USA) were used. Data were expressed as either mean ± standard deviation or median and interquartile range. *p* values of less than 0.05 were considered significant.

## 3. Results

### 3.1. Characteristics of SSc Patients

Our study consisted of two separate cohorts: 36 SSc patients and 26 age- and sex-matched HCs were enrolled in a study of systemic markers of oxidative stress, and 35 SSc patients and 32 age- and sex-matched HCs were enrolled in a study of DNA breaks in leukocytes (Table 1). Most of the participating subjects were women with a mean age in groups ranging from 53 to 61 years. Clinical data from SSc patients (Table 2) show that ATAs were the most prevalent antibodies, followed by ACAs. Among the ATA- and ACA-negative patients, in the first cohort, we identified 2 patients positive for anti-fibrillarin (U3 RNP), 1 for anti NOR-90, 1 for anti-RNA polymerase III, and 1 for anti-Ro52 antibodies; 1 patient was negative for all SSc-related antibodies measured. In the second cohort, we identified 3 patients positive for anti-PM/Scl, 1 for anti-RNA polymerase III, 1 for U1RNP, and 1 for anti-Ro52 antibodies. In the first cohort, the diffuse cutaneous form of SSc predominated, whereas in the second cohort, the limited cutaneous form of SSc predominated. Patients’ SSc involved multiple organ systems. GIT complications were present in the cohort studied for systemic markers of oxidative stress and in the cohort studied for DNA breaks in 61% and 51%, respectively, and interstitial lung disease (ILD) was diagnosed in 42% and 43%, respectively. About half of the patients were treated with immunosuppressants (glucocorticoids, hydrochloroquine, methotrexate, mycophenolate mofetil, cyclophosphamide, and rituximab).

### 3.2. Oxidative Imbalance in the Systemic Circulation of Systemic Sclerosis Patients

To assess oxidative status in SSc patients and HCs, serum ROMs were determined using measured ROOH concentrations, and TAC using Fe^3+^ to Fe^2+^ reducing capacity. ROOH concentrations were similar in both SSc patients and HCs (median (interquartile range (IQR)) 340 (55) U. Carr vs. 322 (89) U. Carr; *p* = ns (Figure 1a). In SSc patients, a greater oxidative stress profile was observed in females compared to men as measured by increased ROM levels in females (343 (54) U. Carr vs. 278 (55) U. Carr, *p* = 0.028). TAC was also above the normal range (reported by the manufacturer) in both SSc patients and HCs but was significantly higher in patients with SSc than HCs (3263 (400) U. Cor vs. 3028 (262) U. Cor, *p* = 0.007) (Figure 1b).

The OSI, combing ROM and TAC values, was significantly increased in SSc patients compared to HCs (65 (26) vs. 50 (25), *p* = 0.0356; normal < 40, borderline 41–65, high 66–120, very high > 121) (Figure 1c). Detailed analysis of OSI values revealed that most SSc patients and HCs were located in quadrant I (high ROMs and high TAC) (Figure 1e). SSc patients were predominantly located in quadrant I and quadrant II (high ROMs, low TAC), whereas HCs were predominantly located in quadrant IV (low ROMs, high TAC). No patient or HC belonged to quadrant III.

### 3.3. Markers of Oxidative Damage to Lipids and DNA Show Similar Levels in Serum from Patients with Systemic Sclerosis and Healthy Controls

We further examined whether oxidative imbalance in SSc patients could cause oxidative damage to macromolecules such as lipids and DNA, and we therefore measured serum levels of 4-HNE as a marker of oxidative damage to lipids and 8-OHdG as a marker of oxidative damage to DNA. Serum levels of 4-HNE (Figure 1f) and 8-OHdG (Figure 1g) were comparable in SSc patients and HCs, although the concentration of 8-OHdG was significantly higher in women than in men (3.12 µg/L (1.03) U. Carr vs. 4.60 µg/L (1.51) U. Carr, *p* = 0.021).

### 3.4. Total Antioxidant Capacity and DNA Oxidation Marker 8-Hydroxy-2-Deoxyguanosine in Serum from Patients with Systemic Sclerosis Are Associated with Lung and Gastrointestinal Complications

To better understand whether oxidative stress might be associated with organ-specific manifestations in SSc patients, we correlated oxidative measurements with the clinical patient data. TAC was significantly increased in SSc patients’ ILD compared to HCs and SSc patients without ILD. Additionally, patients with GIT involvement manifested with increased TAC compared to HCs (Figure 2a). The opposite trend was observed for the marker of DNA oxidation, 8-OHdG, which was reduced in SSc patients with ILD (Figure 2b).

### 3.5. Leukocytes from Patients with Systemic Sclerosis Are Characterized by Increased DNA Breaks and Poorer DNA Repair Mechanisms

Although we could not detect systemic DNA damage in SSc patients using serum 8-OHdG levels, we wanted to assess DNA damage, more specifically DNA breaks, at the cellular level as well (Figure 3a). The number of DNA breaks in leukocytes was quantified as a COMET score, which was significantly higher in SSc patients compared to HCs at baseline, after H_2_O_2_-induced oxidative stress and after H_2_O_2_-induced oxidative stress plus DNA repair mechanisms (Figure 3b).

We also examined how the COMET score changed between the three conditions studied: baseline, after H_2_O_2_ treatment, and after DNA repair. In HCs, the COMET score increased by an average of 21% after H_2_O_2_ treatment compared with the baseline and then decreased by an average of 8% after DNA repair mechanisms (mean (SD): Baseline 88.2 (69.3); after H_2_O_2_ 107.1 (70.1); after DNA repair 98.4 (64.4)). In contrast, in SSc patients, the COMET score increased by an average of 19% after treatment with H_2_O_2_ compared with the baseline and then decreased by an average of 1% after DNA repair mechanisms (mean (SD): Baseline 134.1 (50.7); after H_2_O_2_ 160.0 (58.2); after DNA repair 158.7 (62.4)). This suggests that the repair of DNA breaks after oxidative damage is the affected part of oxidative balance in SSc leukocytes.

### 3.6. Total Antioxidant Capacity and DNA Breaks in Leukocytes from SSc Patients Are Associated with the Presence of Autoantibodies

To investigate whether autoimmunity in SSc patients is related to oxidative status, we compared the levels of various markers of oxidative stress in patients positive for different SSc-specific autoantibodies.

We found that TAC was significantly increased in ATA-positive SSc patients compared to HCs, while higher values were also observed compared to ACA-positive patients and patients negative for ATAs and ACAs (Figure 4a). Among ATA- and ACA-negative SSc patients, TAC was highest in those negative for all other tested SSc-associated autoantibodies, while similar values as in HCs were observed in anti-fibrillarin, anti NOR-90, and anti-Ro-52 autoantibody-positive patients.

The COMET score, indicating the number of DNA breaks in leukocytes, did not differ between the presence of either ATAs or ACAs. However, more DNA breaks seemed to be observed in ATA- and ACA-negative SSc patients (Figure 4b). Among ATA- and ACA-negative SSc patients, the highest COMET scores were in three patients positive for anti-PM/Scl antibodies, and the lowest in one patient positive for anti-RNA polymerase III antibodies, although the score was still above the median value of ACA- or ATA-positive patients.

### 3.7. Pharmacotherapy with Immunosuppressants Is Associated with Greater Systemic Lipid Peroxidation and More DNA Breaks in Leukocytes in Patients with Systemic Sclerosis

We analyzed whether pharmacotherapy with immunosuppressants is associated with oxidative status and oxidative damage to macromolecules in SSc patients. Patients receiving immunosuppressants had similar oxidative lipid damage to HCs, while those without immunosuppressive therapy exerted lower levels (Figure 5a). DNA breaks in leukocytes, expressed as COMET scores, were higher in patients receiving immunosuppressive therapy than in patients without immunosuppressive therapy and HCs (Figure 5b). Trends in COMET scores suggest highest DNA damage in patients on methotrexate and lowest DNA damage in patients on mycophenolate mofetil; however, a larger cohort would be needed to confirm this.

## 4. Discussion

Reports on the presence of oxidative stress in SSc patients are unanimous, linking it to all major processes involved in pathogenesis of SSc, and more specifically to fibrotic processes [16], autoimmunity, inflammation [2], and cellular senescence [4], in most cases because of increased ROMs and/or decreased antioxidants. In our study, we show that oxidative stress is present in patients with SSc, which was confirmed by a significantly increased OSI value in the systemic circulation of SSc patients compared to HCs (Figure 1c). The OSI value indicates a deviation from the normal oxidative balance between oxidant and antioxidant components, as represented by the concentration of ROMs and TAC, respectively (Figure 1d). In our case, OSI was elevated in most subjects due to both high ROM concentration and high TAC (Figure 1e) [2,4,16].

SSc patients and age-matched HCs in our study had similar ROM (hydroperoxide) values, which were increased based on reference values (Figure 1a). In contrast, a 2017 meta-analysis that included three studies showed that hydroperoxide concentrations were higher in SSc patients than in HCs [3]. The differences may be due to the fact that the study by Bourji et al. included patients with more active and severe disease, composed of younger diffuse cutaneous SSc, a more severe form of SSc. In addition, their population was on average 15 years younger [17] than ours, where older age (around 60 years) might already contribute to the elevated ROM values and mask the contribution of SSc itself. It is well known that the elderly population is more susceptible to the development of oxidative stress with increased hydroperoxide levels and lower antioxidant protection [18]. The studies by Riccieri et al. and Firuzi et al. included HCs who were free of acute and chronic inflammatory diseases and pharmacotherapy [19,20]; unfortunately, these criteria could not be met in our elderly cohort. However, in contrast to our study, the ages of the HCs in their studies were not comparable to the ages of the patients, which is known to have a significant impact on ROM values. Moreover, we found that the concentration of hydroperoxides was significantly higher in women, which is consistent with previous findings that oxidative balance is disturbed in menopausal and postmenopausal women, probably due to lower estrogen levels [18]. The net state of ROM depends not only on formation but also on removal, in which antioxidants are involved [21].

Serum TAC was significantly elevated in SSc patients compared to HCs, although values were elevated on average in both groups based on reference values (Figure 1b). TAC was associated with ILD- and GIT-involvement in SSc patients in our cohort (Figure 2a). Elevated TAC could represent an adaptive response of the organism to the disease itself, or a response to increased ROMs [22]. Increased synthesis of antioxidant enzymes and endogenous molecules with antioxidant activity, such as ceruloplasmin and glutathione, has been reported as a response to elevated ROM levels [20]. However, excessive TAC is not necessarily beneficial, as antioxidants do not discriminate between harmful and biologically important radicals. It has been shown that antioxidants, when taken in excess, can also act as oxidants [23]. Studies of TAC in SSc patients are inconclusive: some report increased TAC in patients with SSc [19,20,24], as we did; others report decreased TAC in SSc patients [25,26], and a 2017 meta-analysis concluded that there were no statistically significant differences in TAC between SSc patients and HCs [3]. Because TAC measures both the enzymatic and non-enzymatic antioxidant systems, it provides good information about overall antioxidant status. However, it should be kept in mind that TAC can vary considerably due to a number of external factors such as diet, lifestyle, genetics, metabolism, microbiome, presence of inflammation, etc. [27] and therefore does not always reflect disease alone. At the same time, TAC does not indicate which antioxidant components are altered and in what way. For example, decreased levels of superoxide dismutase and vitamin C have been detected in the serum/plasma of SSc patients, but there are mixed reports regarding other antioxidants (CAT, vitamin E, thiol, etc.) [3].

Systemic oxidative stress in SSc patients was not reflected in oxidative damage to macromolecules such as lipids, as serum concentrations of 4-HNE, one of the end products of lipid peroxidation (Figure 1f), were similar in SSc patients and HCs. Although not previously described in SSc patients, 4-HNE was detected in 1997 in morphea in children, where no altered levels were found [28]. In contrast, other lipid peroxidation markers suggest oxidative lipid damage in SSc patients: increased serum and plasma concentrations of MDA have been reported, in addition to associations of F2-IsoPs with the fibrotic phenotype of the disease and abnormal lipid oxidation of the erythrocyte membrane [16].

Similarly, we did not detect excessive systemic oxidative DNA damage in the serum of SSc patients compared to HCs based on measured 8-OHdG (Figure 1g). However, concentrations were statistically significantly increased in women, which may be due to the increased ROM levels mentioned earlier. Lower 8-OHdG concentrations were seen in patients with ILD (Figure 2b), which may be due to the increased potentially protective TAC in these patients, which can attenuate oxidative DNA damage. Serum levels of 8-OHdG have not yet been described in SSc patients, but it has been measured in urine, where elevated levels were found, which, in contrast to our results, were positively associated with the presence of lung fibrosis, decreased forced vital capacity, and decreased DLCO to alveolar volume ratio [29]. There is a positive correlation between plasma and urine 8-OHdG, but plasma 8-OHdG is considered more sensitive because it also correlates with the presence of several factors such as age, sex, and smoking [30]. Although there are several markers of oxidative DNA damage, 8-OHdG, which is formed when hydroxyl radicals attack deoxyguanosine residues, is considered one of the most commonly used markers of oxidative DNA damage [31].

The systemic oxidative state of an organism does not necessarily reflect the state within individual cells. For this reason, we examined DNA damage, more specifically DNA breaks, in leukocytes using the COMET assay and found more DNA breaks in leukocytes from SSc patients compared with HCs (Figure 3b). This confirms the findings of Vlachogiannis et al., who used the same method to demonstrate a higher presence of endogenous DNA damage in peripheral blood mononuclear cells from SSc patients, especially in patients with diffuse SSc [8]. We found that causing oxidative stress by adding H_2_O_2_ to cells resulted in similar additional DNA damage in SSc and HC leukocytes. However, the ability to repair DNA damage via endogenous mechanisms in cells was impaired in SSc patients compared to HCs, consistent with previously reported results [8]. In peripheral blood mononuclear cells of SSc patients, increased DNA damage is associated with the presence of polymorphisms in DNA repair genes such as XRCC1 and XRCC4 [7]. At the same time, it has been shown that the impaired DNA damage response/repair in peripheral blood mononuclear cells in SSc patients can activate the type I interferon signaling pathway and thus contribute to the fibrosis process [8].

We found an association between oxidative status in SSc and the presence of SSc-related autoantibodies. TAC was elevated in the serum of SSc patients positive for ATAs compared to patients positive for ACAs or negative for both ATAs and ACAs (Figure 4a). ATAs are commonly associated with the diffuse cutaneous form of SSc, which is related to poorer prognosis, more rapid disease progression, and ILD [32]. Increased TAC is associated with ILD and GIT complications in SSc patients, and thus it appears that in SSc patients with poorer prognosis characterized by the presence of ACAs, TAC may indeed represent a defense mechanism in response to disease severity and/or increased ROMs [22]. The question of whether autoantibodies are a cause, effect, or bystander of oxidative imbalance remains open. It has been shown that autoantibodies found in SSc patients can induce ROMs. For example, anti-PDGF receptor antibodies can activate the PDGF receptor on fibroblasts, triggering MAP kinase signaling pathways that lead to excessive production of ROMs, which in turn activate myofibroblasts via increased phosphorylation of ERK and accelerate collagen formation [2]. On the other hand, it has been shown that ROMs can also stimulate autoantibody formation. A study in mice showed that oxidation of topoisomerase I by ROMs was sufficient to cause loss of tolerance to the antigen, leading to production of ATAs and increased fibroblast proliferation and gene expression of type I collagen [33]. In addition, we detected more DNA breaks in the leukocytes of SSc patients who did not have ATAs and ACAs, but surprisingly, we found no differences between ATA-positive and ACA-positive patients (Figure 4b). Although the most common SSc-specific antibodies are directed against DNA topoisomerase I (ATA) and the centromeric proteins CENP-A, B, and C (ACA), other autoantibodies such as RNA polymerase III, U3 RNP complex, Th/To, PM/Scl, U1 RNP, Ku, RuvBL1/2, Ro52, NOR90, and ANP32A are also found in patients with SSc [11]. Additional testing in SSc patients negative for ATAs and ACAs revealed that most DNA breaks occurred in patients positive for anti-PM/Scl antibodies. Up to 20% of SSc patients have an overlap syndrome with other autoimmune diseases, and anti-PM/Scl antibodies are common in patients with overlapping myositis [11]. Anti-PM/Scl antibodies in SSc patients predict a milder disease course without severe organ involvement and a better prognosis in general [34,35], so it is unexpected that these patients had more DNA damage in leukocytes.

Pharmacotherapy can interfere with the oxidative status in the organism as well. In our study, we observed that immunosuppressant therapy was associated with greater systemic oxidative lipid damage (Figure 5a) and more DNA breaks in leukocytes (Figure 5b). In this context, the most lipid damage was observed in patients receiving rituximab (of note, among sera measured for lipid damage there were no patients receiving methotrexate), whereas the most DNA breaks were observed in patients receiving methotrexate. Methotrexate is known to significantly increase lipid peroxidation (MDA) and NO in the liver of rats and to decrease antioxidants such as glutathione, catalase, glutathione peroxidase, and superoxide dismutase [36]. Methotrexate inhibits, among other mechanisms, nicotinamide adenosine diphosphate in the cytosol, which is required for the action of glutathione reductase, which allows glutathione to function [37]. Increased DNA breaks in peripheral blood mononuclear cells in SSc patients have also been associated with the use of the cytostatic drug cyclophosphamide [8], which we also observed. Some immunosuppressants, such as cyclosporine and tacrolimus at normal therapeutic doses, have been shown to inhibit DNA repair [38]. However, we cannot say with certainty that immunosuppressants increase oxidative stress in our patients, because such therapy is usually prescribed for more severe cases of SSc [11], and the increased oxidative lipid damage and DNA breaks that we observe could also reflect a more severe form of the disease; thus, multivariable analysis on larger patient cohort would be needed.

However, although we confirm that oxidative stress is present in patients with SSc, it is difficult to draw specific conclusions about oxidative status and oxidative damage based on single markers. There are still no universal criteria for the assessment of oxidative stress, and different markers for the same condition often do not correlate with each other. The results of different studies are difficult to compare because of the use of different samples (serum, plasma), sample handling (fresh/frozen samples), and variations in detection methods lacking standardization. We are also limited as we detect secondary oxidative stress products, which do not allow us to predict with certainty what exactly is happening at the site of formation of ROMs. The oxidative system is extremely sensitive to a variety of factors, so data on the influence of age, sex, and the stability of measured markers in samples over time would be welcome. Overall, all methods used to assess ROSs are also susceptible to artefact, and appropriate controls are required to be certain of the species and amounts measured [21,23].

## 5. Conclusions

In conclusion, we report that increased TAC observed has been in SSc patients, specifically in those with ILD and GIT involvement. Besides associations of TAC with fibrotic symptoms, we also report association of TAC with autoimmunity, as highest levels were observed in ATA-positive patients.

Although there has been increased interest in development of antioxidant therapies for lung fibrosis and gastroesophageal reflux disease, they have limited effects [39,40]. This stresses the importance of balance in tissue to avoid oxidative stress, where even too much antioxidant might be harmful. Further studies are needed to specify which parts of the antioxidant system in SSc patients are most affected and whether they could be selectively modified.

## Figures and Tables

**Figure 1 biomedicines-11-02110-f001:**
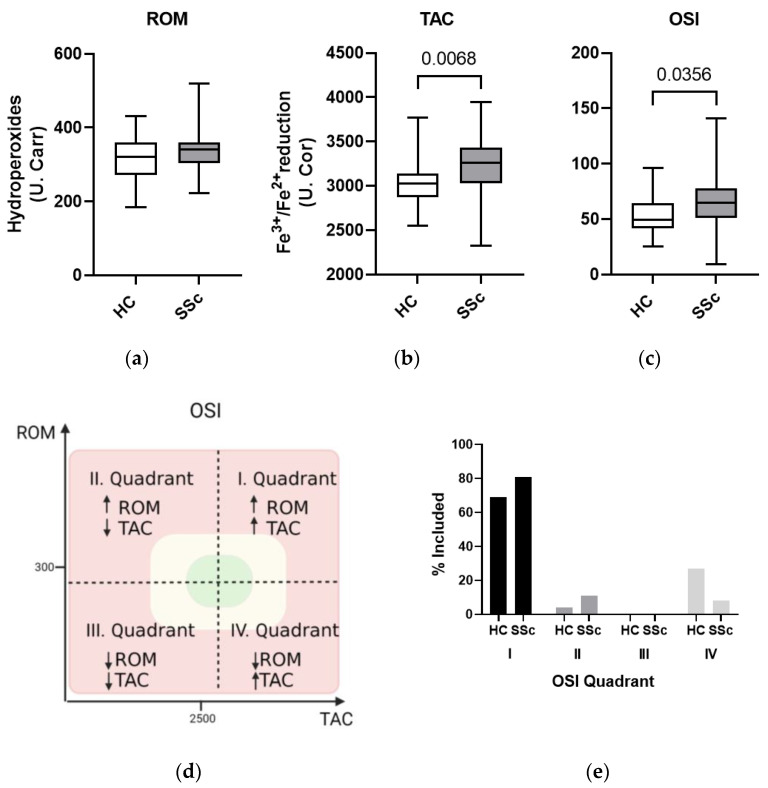
Markers of oxidative stress measured in serum of healthy controls (HCs) and systemic sclerosis (SSc) patients. (**a**) Reactive oxygen metabolite (ROM) concentration, based on measured hydroperoxides, is elevated but similar in both HCs and patients with SSc. (**b**) Total antioxidant capacity (TAC), based on the ability to reduce Fe^3+^ to Fe^2+^, is elevated in both HCs and SSc patients but is significantly higher in SSc patients than HCs. (**c**) the oxidative stress index (OSI) calculated from the values of ROMs and TAC is significantly higher in SSc patients than in HCs. (**d**) Values of OSI divided into four quadrants based on the values of ROMs (*y* axis) and TAC (*x* axis). Arrows represent increase/decrease of ROM/TAC values in quadrants I-IV. Green indicates normal OSI values (0–40), yellow borderline (41–65), and red high (>66). (**e**) Percentage of HCs and SSc patients in each OSI quadrant. (**f**) Oxidative lipid damage (lipid peroxidation) based on 4-HNE levels is similar in HCs and SSc patients. (**g**) Oxidative DNA damage measured using 8-OHdG levels is similar in HCs and SSc patients. Legend: 4-HNE—4-hydroxynonenal; 8-OHdG—8-hydroxy-2-deoxyguanosine; HCs—healthy controls; OSI—oxidative stress index; SSc—systemic sclerosis.

**Figure 2 biomedicines-11-02110-f002:**
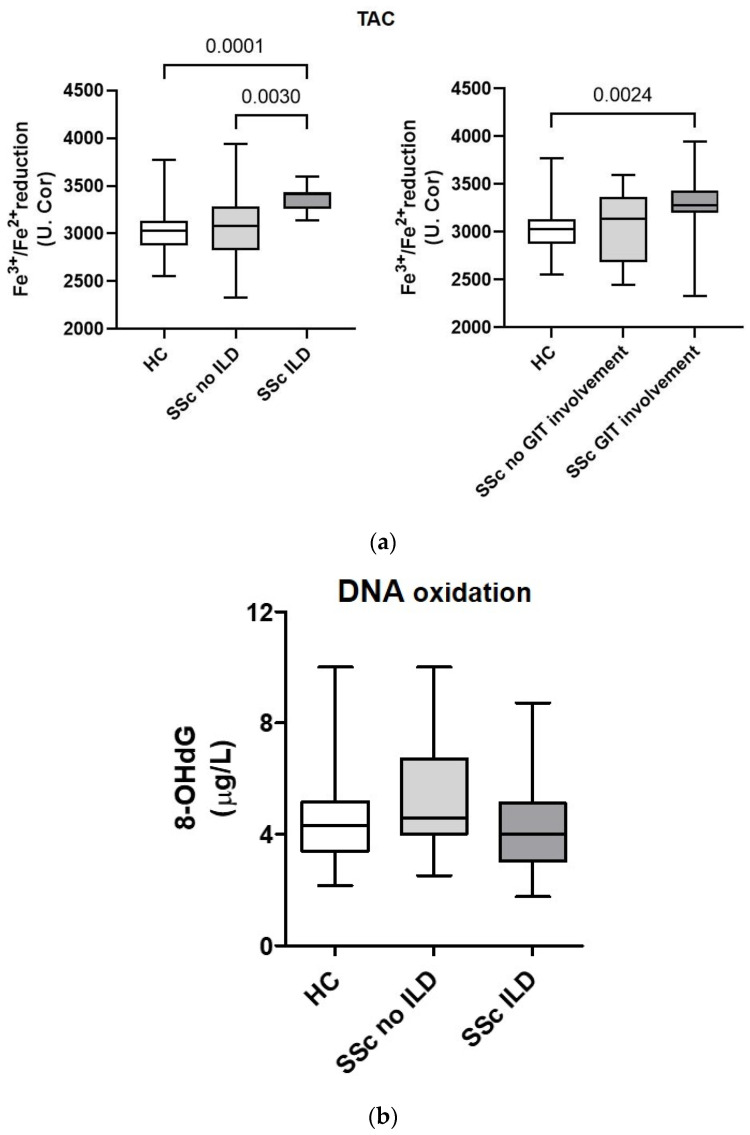
Total antioxidant capacity (TAC) and DNA oxidation measured in serum are related to the clinical manifestations of systemic sclerosis (SSc). (**a**) TAC, based on the ability to reduce Fe^3+^ to Fe^2+^, is increased in SSc patients with interstitial lung disease (ILD) and gastrointestinal tract (GIT) involvement. (**b**) Oxidative DNA damage, based on 8-OHdG levels, is less pronounced in SSc patients with ILD. Legend: 8-OHdG—8-hydroxy-2-deoxyguanosine; GIT—gastrointestinal tract; HCs—healthy controls; ILD—interstitial lung disease; SSc—systemic sclerosis.

**Figure 3 biomedicines-11-02110-f003:**
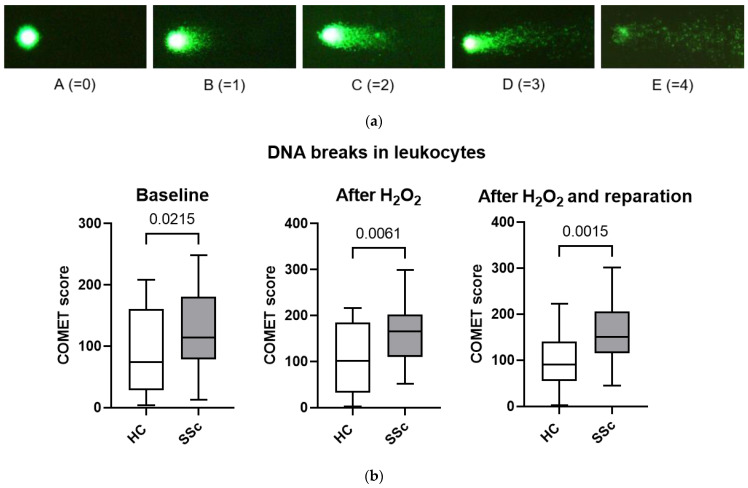
DNA breaks in leukocytes from healthy controls (HCs) and systemic sclerosis (SSc) patients measured using the COMET assay. (**a**) COMET categories A–E as seen under the microscope, where category A represents intact DNA, and category E represents completely damaged DNA. (**b**) COMET scores at baseline, after H_2_O_2_-induced oxidative damage and after H_2_O_2_-induced oxidative damage plus DNA repair mechanisms, are significantly higher in SSc patients than in HCs. A higher COMET score presents more DNA breaks and thus more DNA damage. Legend: H_2_O_2_—hydrogen peroxide; HC—healthy controls; SSc—systemic sclerosis.

**Figure 4 biomedicines-11-02110-f004:**
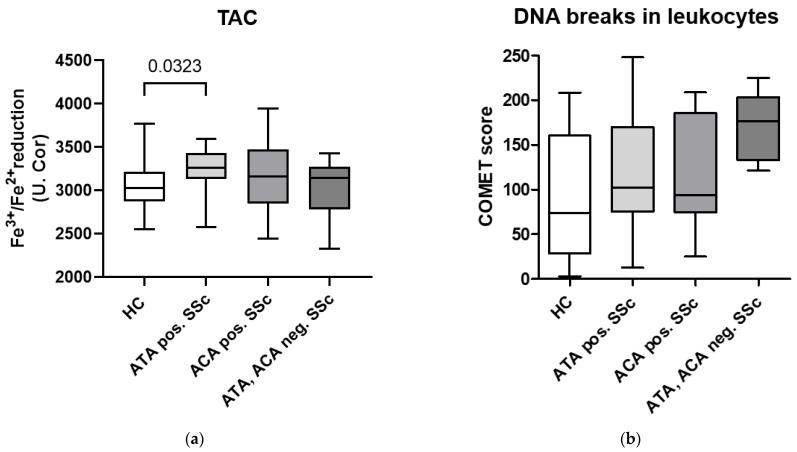
TAC and DNA breaks in leukocytes differ in SSc patients with different autoantibodies. (**a**) TAC, based on the ability to reduce Fe^3+^ to Fe^2+^, is highest in SSc patients positive for ATAs. (**b**) The COMET score, indicating the number of DNA breaks, is highest in SSc patients negative for ATAs and ACAs. Legend: ACAs—anti-centromere antibodies; ATAs—anti-topoisomerase I antibodies; HCs—healthy controls; SSc—systemic sclerosis, TAC—total antioxidant capacity.

**Figure 5 biomedicines-11-02110-f005:**
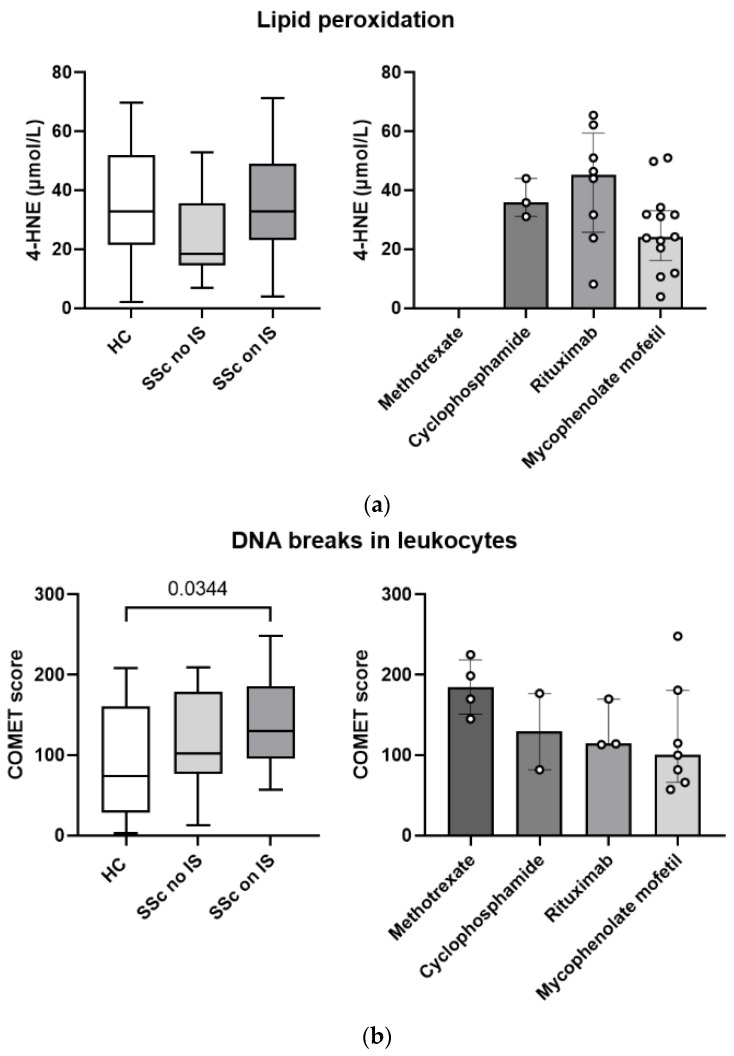
Pharmacotherapy with immunosuppressants (ISs) in patients with systemic sclerosis (SSc) is associated with greater systemic lipid peroxidation and more DNA breaks in leukocytes. (**a**) Lipid peroxidation, based on serum 4-HNE levels, was higher in patients receiving ISs than in patients not receiving ISs and highest in patients receiving rituximab. (**b**) The COMET score, indicating the number of DNA breaks, was higher in leukocytes from patients who received ISs than in leukocytes from patients who did not receive ISs and was highest in leukocytes from patients who received methotrexate. Legend: HCs—healthy controls; ISs—immunosuppressants; SSc—systemic sclerosis.

**Table 1 biomedicines-11-02110-t001:** SSc patients and healthy controls (HCs) enrolled in a study measuring systemic markers of oxidative stress and in a study measuring DNA breaks in leukocytes.

	Systemic Markers of Oxidative Stress		DNA Breaks in Leukocytes	
	SSc	HC	SSc	HC
N (%)	36 (58)	26 (42)	35 (52)	32 (48)
Age (years)				
Median (IQR)	63 (19)	59 (13)	57 (19)	54 (16)
Mean (SD)	61 (14)	58 (13)	56 (13)	53 (13)
Minimum	31	29	29	27
Maximum	88	86	80	84
Sex N (%)				
Male	4 (11)	3 (12)	6 (17)	4 (13)
Female	32 (89)	23 (88)	29 (83)	28 (87)

Legend: HC—healthy controls; IQR—interquartile range; SD—standard deviation; SSc—systemic sclerosis.

**Table 2 biomedicines-11-02110-t002:** Clinical data and laboratory parameters of SSc patients enrolled in a study measuring systemic markers of oxidative stress and a study measuring DNA breaks in leukocytes.

	Systemic Markers of Oxidative Stress N (%)	DNA Breaks in Leukocytes N (%)
SSc clinical subset		
Diffuse cutaneous	20 (56)	11 (31)
Limited cutaneous	14 (39)	22 (63)
No data	1 (3)	2 (6)
Clinical manifestations		
Raynaud’s phenomenon	32 (89)	31 (89)
History of digital ulcers	12 (33)	18 (51)
Digital ulcers	3 (8)	7 (20)
Digital pitting scars	5 (14)	7 (20)
Telangiectasia	18 (50)	21 (60)
Calcinosis	2 (6)	7 (20)
GIT involvement	22 (61)	18 (51)
Interstitial lung disease	15 (42)	15 (43)
Pulmonary arterial hypertension	3 (8)	1 (3)
Autoantibodies		
ACA	10 (28)	14 (40)
ATA	20 (55)	15 (43)
ACA, ATA neg.	6 (17)	6 (17)
Nail fold capillaroscopy pattern		
Early	9 (25)	3 (9)
Active	14 (39)	20 (57)
Late	6 (17)	4 (11)
No data	7 (19)	8 (31)
Comorbidities		
Arterial hypertension	10 (28)	12 (34)
Hyperlipidemia	8 (22)	9 (26)
Type 2 diabetes	4 (11)	1 (3)
Asthma	1 (3)	3 (9)
Chronic obstructive pulmonary disease	0 (0)	2 (6)
Cancer	3 (8)	3 (9)
Coronary artery disease	2 (6)	3 (9)
Atherosclerosis	6 (17)	2 (6)
Treatment		
Immunosuppressants	20 (56)	16 (46)
Glucocorticoids	3 (8)	7 (20)
Methotrexate	0 (0)	4 (11)
Cyclophosphamide	3 (8)	2 (6)
Mycophenolate mofetil	13 (36)	7 (20)
Rituximab	8 (22)	3 (9)
Analgesics and anti-inflammatory drugs	14 (39)	11 (31)
Calcium channel blockers	20 (56)	28 (51)
Prostacyclins	4 (11)	2 (6)
PDE5 inhibitors	6 (17)	5 (14)
Laboratory parameters	Median (IQR)	Median (IQR)
Leukocytes (×10^9^/L)	66 (49)	64 (28)
Thrombocytes (×10^9^/L)	251 (99)	232 (87)
Hemoglobin (g/L)	134 (16)	131 (17)
SR (mm/h)	18 (19)	20 (26)
IL-6 (ng/L)	4 (6)	2 (3)
Pulmonary function	Median (IQR)	Median (IQR)
DLCO (%)	73 (28)	70 (30)
FVC (%)	98 (22)	91 (27)
FEV1 (%)	95 (21)	92 (21)

Legend: ACAs—anti-centromere antibodies; ATAs—anti-topoisomerase I antibodies; DLCO—diffusing capacity for carbon monoxide; FEV1—forced expiratory volume in 1 s; FVC—forced vital capacity; GIT—gastrointestinal tract; IL-6—interleukin-6; IQR—interquartile range; PDE5—phosphodiesterase type 5; SR—sedimentation rate; SSc—systemic sclerosis.

## Data Availability

The data presented in this study are available on request from the corresponding author.

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
