# Peer review of "Disturbed Antioxidant Capacity in Patients with Systemic Sclerosis Associates with Lung and Gastrointestinal Symptoms"

_biomedicines, 2023, doi:10.3390/biomedicines11082110_

Round 1
Reviewer 1 Report
The Authors analized in this study the oxidative stress level (ROM) and antioxidant capacity (TAC) firstly. In the second phase the lipid and DNA damage was directly investigated as a second cohort. These investigations are not in close relation to each other, the Authors may mull to publish the results separately. The nearly significant results suggests that the study is mildly underpowered, but the SSc is considered to be a rare disorder. There are some major issues to be clarified and answered:
1. The normal values of ROM and TAC must be interpreted in the methods.
2. The calculation of OSI must be exactly defined.
3. I suggest to create a table which contains the patient’s absolute number and percents in SSc and control group with isolated ROM, isolated TAC and double abnormality (values above and below normal range, and both above and below normal range, respectively). Otherwise the OSI quadrants may not easily analized.
4. The main weakness of the first part of this study is the high rate of ROM and TAC abnormality in the healthy control group. There is a sentence in Discussion to be explained: „The studies by Riccieri et al. and Firuzi et al. included HC who were free of acute and chronic inflammatory diseases and pharmacotherapy (19, 20); unfortunately, these criteria could not be met in our elderly cohort.”. The Healthy Control group contains patients with acute or chronic illnesses? It must be presented in a table as baseline characteristic of healthy controls. The answere for this question is considered to be the cornerstone of the paper.
Reviewer 2 Report
General comments:
The authors reported a disturbed antioxidant capacity in patients with SSc with relation to organ symptoms. The manuscript was well documented and will provide useful information to readers. Some minor resolutions will be required to publish in the Journal.
Specific comments:
1. As the authors stated, there might be present a delicate imbalance between oxidant and antioxidant components in not only patients with SSc but also measuring them in laboratory procedures. Oxidative chemical reactions would arise in a short minute. They should describe a difficulty in checking both oxidative and antioxidative measuring.
2. Among pathological processes of autoimmunity, inflammation and fibrosis in relation to SSc, how were the oxidative and antioxidative components correlate with pathological processes?
3. How do oxidations associate with autoantibodies synthesis in pathophysiology of SSc and clinical symptoms like lung fibrosis, GI changes, renal crisis, and pulmonary hypertensin, etc?
4. It is important to start treatments in the initial stage of SSc just after diagnosis, are the results of the present study useful to begin the therapy?
Fine.
There are some technical errors.
Reviewer 3 Report
The manuscript entitled "Disturbed antioxidant capacity in patients with systemic sclerosis associates with lung and gastrointestinal symptoms" brings an important inside in mechanism of oxidative stress in SS associated lung and GI symptoms
The following observations have to be made
Introduction
Please rewrite the aim of the study because you are talking about inflammation, autoimmunity (antibodies) and oxidative stress, and after that you are mentioning only the oxidative stress and other parameters. Please make a pathogenic connection between all of these concepts.
Please mention clearly the aim of the study, according with all of these parameters.
Please mention the Ethic Committee Approval and if the patients signed an Informed Consent.
Conclusion has to be a separate chapter according with your Discussions
Round 2
Reviewer 1 Report
The current version of paper can be accepted-